# Assessment of Periprostatic and Subcutaneous Adipose Tissue Lipolysis and Adipocyte Size from Men with Localized Prostate Cancer

**DOI:** 10.3390/cancers12061385

**Published:** 2020-05-28

**Authors:** Dushan Miladinovic, Thomas Cusick, Kate L. Mahon, Anne-Maree Haynes, Colin H. Cortie, Barbara J. Meyer, Phillip D. Stricker, Gary A. Wittert, Lisa M. Butler, Lisa G. Horvath, Andrew J. Hoy

**Affiliations:** 1Discipline of Physiology, School of Medical Sciences, Charles Perkins Centre, Faculty of Medicine and Health, The University of Sydney, New South Wales 2006, Australia; dmil3834@uni.sydney.edu.au; 2Cancer Division, The Kinghorn Cancer Centre/Garvan Institute of Medical Research, New South Wales 2010, Australia; t.cusick@garvan.org.au (T.C.); kate.mahon@lh.org.au (K.L.M.); a.haynes@garvan.org.au (A.-M.H.); pstricker@stvincents.com.au (P.D.S.); lisa.horvath@lh.org.au (L.G.H.); 3Discipline of Medicine, Central Clinical School, The University of Sydney School of Medicine, Faculty of Medicine and Health, The University of Sydney, New South Wales 2006, Australia; 4Department of Medical Oncology, Chris O’Brien Lifehouse, New South Wales 2050, Australia; 5Royal Prince Alfred Hospital, New South Wales 2050, Australia; 6School of Medicine, Lipid Research Centre, Molecular Horizons, University of Wollongong, New South Wales 2522, Australia; colinc@uow.edu.au (C.H.C.); bmeyer@uow.edu.au (B.J.M.); 7Illawarra Medical Research Institute, University of Wollongong, New South Wales 2522, Australia; 8St. Vincent’s Clinical School, The University of New South Wales, New South Wales 2010, Australia; 9St. Vincent’s Prostate Cancer Centre, St. Vincent’s Clinic, New South Wales 2010, Australia; 10South Australian Health and Medical Research Institute, South Australia 5000, Australia; gary.wittert@adelaide.edu.au (G.A.W.); lisa.butler@adelaide.edu.au (L.M.B.); 11School of Medicine and Freemasons Foundation Centre for Men’s Health, University of Adelaide, South Australia 5000, Australia

**Keywords:** prostate cancer, periprostatic adipose tissue, lipolysis, localized prostate cancer, subcutaneous adipose tissue

## Abstract

The prostate is surrounded by periprostatic adipose tissue (PPAT), the thickness of which has been associated with more aggressive prostate cancer (PCa). There are limited data regarding the functional characteristics of PPAT, how it compares to subcutaneous adipose tissue (SAT), and whether in a setting of localized PCa, these traits are altered by obesity or disease aggressiveness. PPAT and SAT were collected from 60 men (age: 42–78 years, BMI: 21.3–35.6 kg/m^2^) undergoing total prostatectomy for PCa. Compared to SAT, adipocytes in PPAT were smaller, had the same basal rates of fatty acid release (lipolysis) yet released less polyunsaturated fatty acid species, and were more sensitive to isoproterenol-stimulated lipolysis. Basal lipolysis of PPAT was increased in men diagnosed with less aggressive PCa (Gleason score (GS) ≤ 3 + 4) compared to men with more aggressive PCa (GS ≥ 4 + 3) but no other measured adipocyte parameters related to PCa aggressiveness. Likewise, there was no difference in PPAT lipid biology between lean and obese men. In conclusion, lipid biological features of PPAT do differ from SAT; however, we did not observe any meaningful difference in ex vivo PPAT biology that is associated with PCa aggressiveness or obesity. As such, our findings do not support a relationship between altered PCa behavior in obese men and the metabolic reprogramming of PPAT.

## 1. Introduction

Prostate cancer (PCa) is the most commonly diagnosed cancer in men in developed countries and the second most common cancer worldwide [1]. Several studies have reported that PCa pathobiology is influenced by obesity, specifically, more aggressive carcinoma, poorer treatment outcomes, and increased cancer-specific mortality in obese men [2,3,4,5,6,7,8,9]. A range of systemic mechanisms has been proposed to underpin this association, including altered adipokine and inflammatory profiles, hyperinsulinemia, and dyslipidemia [10,11,12].

Significant attention has focused on the role that the local adipose bed that surrounds the prostate, called periprostatic adipose tissue (PPAT), plays in influencing PCa biology. For example, higher grade or more aggressive PCa has been linked to greater PPAT thickness (for review, see [13]), adipose infiltrating into the tumor [14], and invasion of PCa into PPAT (defined as an extracapsular extension) [15,16,17]. Additionally, a reciprocal interaction between adipocytes and PCa cells has been reported, both in terms of PPAT stimulating PCa aggressiveness but also PCa cells altering adipocyte biology [18,19,20,21,22]. Direct tumor–adipocyte interaction does occur in vivo in multiple situations. Specifically, the vasculature that services the prostate crosses PPAT, thereby facilitating local tissue crosstalk, at the anterior surface of the prostate, where there is direct contact with PPAT [23], as well as during extracapsular extension of the tumor. As such, it is conceivable that the poorer PCa outcomes observed in obese populations are, in part, due to a paracrine relationship between PPAT and PCa that may be altered in an obese setting. However, there are no existing reports on the PPAT lipid phenotype, including lipolysis, from men with localized PCa, and whether metabolic rewiring of PPAT associates with patients’ metabolic traits and indices of PCa aggressiveness.

In this study, we assess the lipolytic profile and adipocyte size of PPAT and compare it to that of SAT in men undergoing radical prostatectomy for localized PCa. Further, we explore the relationship between adipose lipid biology and metabolic profiles and PCa clinical features. We hypothesize that lipid-attributes of PPAT differ from SAT, and that PPAT lipolysis is increased in men with more aggressive PCa and in obese men.

## 2. Results

### 2.1. Patient Characteristics

The patient cohort was comprised of 60 patients with a mean age of 62 years ± 8 (range: 42 to 78; Table 1) with a mean BMI of 26.9 kg/m^2^ ± 3.4 (range: 21.3 to 35.6), indicating a slightly overweight population. The mean waist circumference (WC) was 95.8 cm ± 8.7 (range: 81 to 117), indicative of a WC linked with an elevated risk of all-cause mortality [24]. Mean fasting blood serum glucose and insulin levels were within the normal range (Table 1), with 1 and 21 patients who had hyperinsulinemia and hyperglycemia, respectively. The cohort contained four patients who had type 2 diabetes at the time of radical prostatectomy surgery, with only one patient using metformin. Furthermore, mean blood serum cholesterol, triacylglycerols, HDL, and LDL for all 60 patients were within the normal range (Table 1), with 14 patients identified as having hypercholesterolemia and no patients having hypertriglyceridemia.

Of the 60 patients assessed, 9 patients were using statin medication. There was no difference in the average age, BMI, and WC between the men taking statins compared to men not taking statins, as expected. There was the expected reduction in circulating cholesterol (No Statin: 4.6 ± 1.0 mM, Statin: 3.8 ± 0.7 mM, *p* = 0.01) and LDL (No Statin: 2.8 ± 0.8 mM, Statin: 1.8 ± 0.9 Mm, *p* = 0.005) in patients taking statins. Blood serum glucose was significantly elevated in statin patients (No Statin: 5.2 ± 0.7 mM, Statin: 5.8 ± 0.7 mM, *p* = 0.05).

### 2.2. PPAT Adipocytes are Smaller than Subcutaneous Adipose Tissue Adipocytes and Secrete Less Polyunsaturated Fatty Acids but Basal and Stimulated Lipolysis are Essentially the Same

It is well established that there are anatomical, cellular, molecular, and physiological differences between SAT and visceral adipose tissue of the abdominal cavity [25]. However, it is unclear whether these differences extend to PPAT. Using quantitative histomorphometry, PPAT adipocytes were smaller than those in SAT from the same patient, as determined by measuring the mean size (Figure 1A) and the distribution of adipocyte sizes (Figure 1B). There was no relationship between mean adipocyte size from PPAT and SAT (R^2^ = 0.0004, *p* = 0.9).

The rate of basal fatty acid release from PPAT and SAT explants was similar (Figure 1C); however, there was no correlation between PPAT and SAT basal lipolysis (Figure 1D). Further, there was no correlation between the basal lipolysis of PPAT and SAT with the age of the patients (PPAT: R^2^ = 0.004, *p* = 0.6; SAT: R^2^ = 0.0009, *p* = 0.8), or with mean adipocyte size (PPAT: R^2^ = 0.006, *p* = 0.7; SAT: R^2^ = 0.03, *p* = 0.4).

Next, we determined the abundance of fatty acid species that were released by PPAT and SAT in the basal state. Of the 28 species measured, 6 species were less abundant in the secretome of PPAT, specifically 15:1 n-10, 18:3 n-6, 20:3 n-6, 20:4 n-6, 22:5 n-3, and 22:6 n-3, and total polyunsaturated fatty acids (PUFAs), both n-6 and n-3 PUFAs, were less abundant compared to SAT (Table 2). Conversely, three species were more abundant in PPAT secretome, specifically 16:0, 20:0, and 22:1 n-9 (Table 2). In general, the changes in the abundance of fatty acid species secreted from PPAT and SAT were small, and so there were little striking differences between these adipose beds.

We next assessed the responsiveness of PPAT and SAT explant lipolysis to hormone stimulation. As expected, insulin suppressed lipolysis, whereas both concentrations of isoproterenol increased lipolysis in both PPAT and SAT explants (Figure 1E and individual data reported in Appendix A). However, PPAT was more sensitive to isoproterenol (1 µM) than SAT (Δ 8.3 pmol/mg/2h, ~9% difference, *p* = 0.0009; Figure 1E). Androgens, such as DHT, are critical drivers of PCa [26] and directly impact adipose lipid metabolism [27]. In this study, DHT did not alter fatty acid release in either PPAT or SAT explants (1 nM; Figure 1E and Appendix A). Importantly, similar patterns were observed when expressing the results as a percentage change relative to basal (Figure 1F). Collectively, these data show that PPAT adipocytes are smaller and secrete less PUFA species than SAT, and basal PPAT lipolysis and its response to extracellular stimuli are effectively the same as SAT and are consistent with other adipose beds [28].

### 2.3. PPAT Adipocyte Size and Lipolysis Does not Differ between Low and High Aggressive Disease

A Gleason score (GS) is used to grade clinical PCa pathology. We dichotomized our cohort into a less aggressive “GS ≤ 3 + 4” and a more aggressive group “GS ≥ 4 + 3” since patients with “≥4 + 3” disease have greater rates of relapse compared to “≤3 + 4” [29]. From the 60 patients collected, 32 patients had less aggressive PCa, and 28 had more aggressive PCa (Table 3). There was no difference in factors that define the metabolic profile of the patients, such as BMI, blood serum, and lipid chemistry between “≤3 + 4” and “≥4 + 3” (Table 3). There was a trend for WC to be greater in the “≥4 + 3” cohort compared to the “≤3 + 4” cohort (*p* = 0.054), whilst PSA and age were greater in the “≥4 + 3” cohort compared to the “≤3 + 4” cohort (Table 3).

There was no difference in the mean size or the distribution of adipocyte sizes between GSs of “≤3 + 4” (less aggressive) and “≥4 + 3” (more aggressive; Figure 2A,B). Somewhat surprisingly, PPAT explants from patients with more aggressive PCa had lower rates of basal fatty acid release (~9% reduction) compared to less aggressive disease (Figure 2C). This slight reduction in total fatty acid release was not due to changes in the abundance of specific fatty acid species, except for 24:1 n-9, which constituted 0.03% ± 0.02% of fatty acids secreted by PPAT from the more aggressive disease compared to 0.11% ± 0.03% from less aggressive PCa (Table 4). Others have reported that PPAT from men with more aggressive disease has elevated intracellular monounsaturated and reduced saturated fatty acid levels relative to SAT [30]. However, we did not see any differences in total saturated, monounsaturated, and polyunsaturated fatty acids released from PPAT (Table 4).

Despite the slight reduction in the basal release of fatty acids from PPAT obtained from more aggressive PCa, there was no difference in response to insulin, isoproterenol, or DHT stimulation between GS “≤3 + 4” and “≥4 + 3” patients (Figure 2D). Likewise, we did not detect differences in PPAT adipocyte size and lipolysis rates when dichotomizing the cohort using other clinical features, including positive extracapsular extension (Appendix A) and positive lymph nodes. As such, PPAT adipocyte size and ex vivo lipolysis are not influenced by PCa disease aggressiveness.

SAT fatty acid composition has previously been reported to differ between PCa and benign hyperplasia patients [31]. Specifically, PCa patients had higher SAT 18:0, 22:5 n-3, and saturated fatty acids, and lower 18:1 n-9, monounsaturated fatty acids, and ratios of monounsaturated to saturated fatty acids than benign hyperplasia patients [31]. However, in the present study, we did not observe any major differences in the profile of fatty acid species secreted by SAT fatty acid in men with less or more aggressive disease.

Waist circumference (WC) is a better predictor of all caused mortality than BMI in populations ≥50 years old [32,33], and individuals with a WC ≥ 95 cm have an elevated risk of all-cause mortality [24,34]. As such, we dichotomized the cohort by WC into those with a normal WC (i.e., <95 cm) and those with a large WC (i.e., ≥95 cm). Only 50 of the 60 patients in our cohort had their WC recorded. From these 50 patients, 24 had normal WC (<95 cm), and 26 had high WC (≥95 cm; Table 5). As expected, BMI was greater in men with high WC compared to low WC (*p* = 0.0005; Table 5). Cholesterol was elevated (*p* = 0.04) in men with high WC compared to men with low WC, with LDL (*p* = 0.07) trends being elevated in patients with a high WC (Table 5). There were no differences in insulin levels (*p* = 0.13). PSA (*p* = 0.74) and age (*p* = 0.46) were not different between patients with normal or high WC, as expected (Table 5).

Mean PPAT adipocyte size, and the distribution of adipocyte sizes, was not different between low and high WC patients (Figure 3A,B). Likewise, there was also no difference in basal PPAT lipolysis (Figure 3C), the abundance of fatty acid species released from PPAT explants (Table 6), or the responsiveness of PPAT lipolysis to hormonal stimulation (Figure 3D). SAT mean adipocyte size, and basal- and hormone-stimulated lipolysis were similar for low and high WC patients (Appendix A).

There was no difference in the mean size of PPAT adipocytes in patients who are overweight or obese, as defined by BMI (*p* = 0.13; Figure 3E), or in the distribution of adipocyte sizes (Figure 3F). Further, there was no difference in basal- (Figure 3G) and hormone-stimulated fatty acid releases (Figure 3H), or the abundance of fatty acid species (Table 6) from PPAT explants. PPAT of obese men had increased intracellular monounsaturated/saturated lipid ratios compared to subcutaneous fat [35,36]. However, we did not observe any differences in the total amount of saturated, monounsaturated or polyunsaturated fatty acids (Table 6), nor in the monounsaturated/saturated ratios of secreted fatty acids between lean, overweight, and obese groups (Lean: 0.90 ± 0.04; Overweight: 0.86 ± 0.02; Obese: 0.82 ± 0.04, *p* = 0.40 main effect by ANOVA), nor between low and high WC (Low WC: 0.88 ± 0.03; High WC: 0.86 ± 0.03, *p* = 0.7). Finally, we saw similar patterns in SAT (Appendix A). Collectively, these results show that key attributes of adipose lipid biology are not different in men undergoing radical prostatectomy with different WC or BMI classifications.

## 3. Discussion

Several studies have reported that the clinical outcomes of PCa are influenced by obesity, with obese patients more likely to have more aggressive carcinoma, poor treatment outcomes, and increased cancer-specific mortality [7,8]. A hallmark feature of obesity is adipose tissue expansion and altered adipocyte fatty acid-triacylglycerol metabolism [37], and when combined with the reports that the amount of PPAT correlates with higher grade or aggressiveness of disease (for review, see [13]), it is an attractive hypothesis that an adipose–PCa axis underpins these clinical observations in obese patients. Here, we show that PPAT adipocytes are smaller than those adipocytes from the same person’s SAT samples and, despite subtle differences in the fatty acid species profile secreted, there was no difference between PPAT and SAT explants in the amount of fatty acid release on a per-milligram basis. PPAT lipolysis responded as expected to insulin (anti-lipolytic) and isoproterenol (pro-lipolytic); however, PPAT was more sensitive to 1 µM isoproterenol stimulation than SAT explants. Counter to our hypothesis, the basal rate of fatty acid release from PPAT explants was greater in men with less aggressive PCa, yet we observed no changes in the abundance of fatty acid species secreted by PPAT explants nor the rate of hormone-stimulated lipolysis or adipocyte size between men with more or less aggressive disease. Further, there was no difference in the rate of fatty acid release from PPAT explants collected from low or high WC patients. These observations, in an adequately sized cohort of 60 patients, demonstrate that there is no remodeling of PPAT lipolysis or adipocyte size that associates with PCa aggressiveness or metabolic features (i.e., WC, BMI) in men undergoing radical prostatectomy.

There is a growing appreciation that the “host” plays a major role in supporting the establishment and progression of cancer [38,39,40]. A key example is the influence that obesity has on cancer risk and progression. Whereas obesity, as determined by BMI, is not associated with PCa risk [3,41], it is associated with altered PCa progression [2,4]. Specifically, several studies have reported a correlation between BMI and PCa aggressiveness [2,3,4,5,6], while other studies have not [42,43]. We observed no differences in the mean BMI between high-grade PCa disease (GS ≤ 3 + 4) compared to men with low-grade disease (GS ≥ 4 + 3; Table 3); however, our study was not designed to test this, and so we are underpowered for this analysis. In any event, BMI is a flawed measure of adiposity/obesity whereas WC, as a measure of abdominal obesity, has been shown to be a better predictor of obesity-associated all-cause mortality in populations over 50 yrs. [33,44]. Additionally, men with a larger WC (i.e., greater central obesity) have elevated risk of high-grade PCa and PCa mortality [5,6]. In this study of 60 men with an average age of 62 years old, WC was greater in men with high-grade PCa disease (GS ≥ 3 + 4) compared to men with low-grade disease (GS ≤ 4 + 3; Table 3), which is consistent with similar studies of men in this age group with PCa [43,45]. Despite this increase in WC, there was no difference in the levels of circulating factors, including insulin, glucose and lipids, which have been proposed to be key mediators of the effect of obesity on PCa progression (see review in [12]). As such, the role that circulating factors play in the link between obesity and altered PCa progress remains to be elucidated.

Due to its anatomical relationship with the prostate and its suggested role in PCa, PPAT has received significant attention in recent years (for reviews, see [13,46]). To date, there has been limited characterization of PPAT biology, with interleukin-6, pro-MMP-9, and other proteins reported to be secreted from PPAT [21,47,48]. We report here that PPAT has similar attributes to other visceral adipose tissue beds. Specifically, PPAT has smaller adipocytes compare to SAT, which is in line with published observations that SAT adipocytes are larger than omental or mesentery visceral adipose beds [49]. We did not observe any difference in the basal rate of fatty acid release from PPAT compared to SAT, whereas others have reported that omental visceral adipose tissue has higher basal lipolytic rates compared to SAT [50,51]. Further, PPAT was less responsive to the anti-lipolytic effect of insulin compared to SAT, which is consistent with published observations made in other visceral adipose beds [52,53]. Whilst we did not see depot-specific differences in the bulk release of fatty acids from PPAT and SAT, we did observe subtle changes in the profile of fatty acid species released from each depot. Specifically, we observed less PUFA species released from PPAT compared to SAT, including 18:3 n-6, 20:3 n-6, 22:5 n-3, and 22:6 n-3, and increased release of saturated species, such as 16:0 and 20:0. Other studies have assessed the intracellular fatty acid profile of PPAT and SAT from men with PCa [30,31,54] but have not performed a direct comparison between PPAT and SAT. It is important to note that the plasma free fatty acid profiles do not correlate well with the fatty acid composition of adipose tissue [55]. This supports our observations that show that saturated fatty acids are the predominant species secreted by PPAT and SAT, whereas others have shown that monounsaturated fatty acids are the major species of intracellular lipids in PPAT, SAT, and visceral adipose tissue [30,31,54,55,56]. Overall, our results demonstrate that PPAT biology differs from SAT biology and that these differences are similar to reported differences between SAT and other visceral adipose depots [49,50,51]. However, we were not able to collect visceral adipose tissue, and so we are limited to the comparison of PPAT biology to SAT. A visceral adipose tissue control would help elucidate differences between the PPAT and visceral adipose tissue biology to inform on whether the presence of local tumors influences PPAT and not overall visceral adipose tissue biology.

In this study, we observed lower levels of PUFAs (18:3 n-6, 20:3 n-6, 20:4 n-6, 22:5 n-3, 22:6 n-3) being secreted from PPAT explants compared with those secreted from SAT. The reduction in the levels of secreted fatty acid species may be simply due to less of those species being stored within adipocytes of adipose tissue. To date, intracellular PPAT and SAT fatty acid species has not been defined. That said, one study did report that the levels of PUFA 20:4 n-6 extracted from PPAT (i.e., intracellular) obtained from men with PCa were lower than from the control group (benign prostatic hyperplasia) [54]. We did not observe any differences in the levels of PUFAs released from PPAT from men with less or more aggressive disease, but we did not have access to cancer-free men. Interestingly, another study [31] reported that PUFA species (20:3n-6, 20:4n-6, 22:5n-6, 22:5n-3, 22:6n-3) were less abundant within the malignant prostate gland compared to a benign prostatic hyperplasia control group. Based upon these observations (see Appendix A), one could speculate that the lower levels of PUFA in PPAT, in PPAT in men with PCa, and the malignant prostate gland itself are due to these PUFAs being released and converted into prostaglandins, leukotrienes, and related compounds, which was also suggested by Careaga and colleagues [54]. It is well known that prostaglandins, leukotrienes, and related compounds derived from n-6 PUFAs are proinflammatory, whereas n-3 PUFAs are antiinflammatory [57] and can reduce telomere shortening, which is related to aging and chronic diseases [58]. A recent review by Frietas and Campos [59] suggested that there is preclinical evidence that n-3 PUFAs and their metabolites might modulate pivotal pathways underlying complications secondary to cancer. Furthermore, a review of human interventional and observational studies reported that higher n-3 PUFA intakes were associated with decreased PCa mortality, but more research is warranted [60].

Several studies have shown that PPAT-derived signals, including proteins and lipids, can exacerbate PCa progression in vitro [21,48,61]. Complimenting this evidence of paracrine interaction are a number of reports that more aggressive PCa disease is associated with infiltrative adipose tissue into the prostate capsule [14] and greater PPAT thickness, independent of BMI status [42,62]. A key question we set out to answer is whether aspects of ex vivo PPAT lipid biology differ between patients with less or more aggressive PCa. Our rationale was that more aggressive disease (i.e., GS ≥ 4 + 3, extracapsular extension) is associated with increased basal PPAT lipolysis, and thereby smaller adipocytes, due to tumor-derived signaling “rewiring” PPAT [20]. Surprisingly, we show that men diagnosed with less aggressive (GS ≤ 3 + 4) PCa had a rate of basal fatty acid release from PPAT explants at a per-unit level that was 9% greater (~6 pmol/mg/2 h) than men diagnosed with a more aggressive cancer (Figure 2). However, we did not see any change in the rate of basal fatty acid release from PPAT explants from patients with extracapsular extension (Appendix A) or positive lymph nodes. This is important since extracapsular extension is defined as the invasion of PCa into PPAT, resulting in the juxtaposition of PCa cells with cells of the adipose tissue, and patients with this pathological feature have poorer prognostic outcomes [15,16,17,63,64,65,66]. Whilst we report a subtle difference in basal fatty acid release between less and more aggressive disease, there was no difference in the hormone-stimulated PPAT lipolysis. The net effect is that there was no difference in mean adipocyte size or the distribution of adipocyte size in PPAT from men with low- and high-grade PCa. It is important to highlight that adipocyte size is influenced by a range of factors, including extracellular FA availability, pro- and antilipolytic stimulation, and differentiation status [67,68,69]. Since we observed only a minor difference in the total amount of fatty acid released from PPAT during a two-hour incubation, we hypothesized that there might be differences in the composition of fatty acid species released. This is because a study reported that PPAT lipids (i.e., intracellular lipids) were enriched with monounsaturated fatty acids in men with more aggressive disease [30]. However, another study reported that there were only minor differences in the fatty acid composition of PPAT from patients undergoing radical prostatectomy for localized prostate tumors and patients undergoing adenomectomy for benign prostatic hyperplasia [30,54]. We saw no differences in the profile of fatty acid species released from PPAT from men with low- or high-grade PCa undergoing radical prostatectomy surgery. As such, we conclude that there is no “rewiring” of PPAT lipolysis that associates with high-grade disease. However, this does not preclude the potential for differences in fatty acid release in vivo where the PPAT will be under hormonal stimulation, which fluctuates dependent upon physiological responses to the environment [70,71], as well as tumor-derived secreted factors that likely influence PPAT lipid biology [19].

Like many other cancer types (see reviews in [72,73]), PCa biology is influenced by obesity [13]. PCa that exists in an obese host is associated with more aggressive disease and increased cancer-specific mortality [2,3,4,5,6,7,8]. Obesity is defined by expansion and alterations in the biology of adipose tissue, including infiltration and activation of immune cells, larger adipocytes, and altered adipocytokine and adipokine release, commonly referred to as adipose tissue dysfunction. This dysfunction in adipose biology has been proposed to be a key driver of altered cancer progression, including PCa (see reviews [46,74]). As such, we hypothesized that PPAT lipid biology would be altered in obese men, as defined by a large WC or BMI. Counter to our hypothesis, we observed no differences in adipocyte size, basal- or hormone-stimulated lipolysis, or in fatty acid species abundance when dividing the cohort by WC or BMI. We did observe that men with more aggressive localized disease (GS ≥ 4 + 3) showed a trend to have a greater mean WC compared to men with less aggressive disease (GS ≤ 3 + 4; *p* = 0.054), but there was no difference in mean BMI. Whilst we did not see changes in PPAT lipid biology in lean or obese men, this does not negate the role of other aspects of PPAT biology that may influence PCa aggressiveness. For example, the gene expression profile of PPAT from obese men favors hypercellularity and reduced immune-surveillance [75], as well as having higher MMP9 activity [20] and increased expression of the chemokine CXCL1 [76]. PPAT from obese men exhibits increased angiogenic capacity, and conditioned media generated from PPAT from obese subjects promotes PC3 PCa cell line proliferation more than conditioned media from lean subjects [35]. Likewise, PC-3 invasion was increased following co-cultivation with adipose tissue from high-fat-diet obese mice compared to cells cultured with adipose tissue from low-fat-diet lean mice [19]. Importantly, others have shown that there was no difference in patient-derived xenografts of moderate-grade localized PCa tumorigenicity in lean and high-fat diet-induced obese severe combined immunodeficiency (SCID) mice [77]. It is important to note that PPAT thickness, as measured by MRI or similar techniques, does not correlate with BMI [78] and so the relationship between patient obesity, PPAT, and PCa biology remains complex and unresolved.

## 4. Materials and Methods

### 4.1. Tissue Collection and Clinical Data

Anterior PPAT and abdominal SAT were removed and collected from men that were diagnosed with localized PCa, who opted for radical prostatectomy via a robotic da Vinci technique at St Vincent’s Hospital. Clinical data of the patients, prostate, and PCa was obtained from the APCRC-NSW/Australian Prostate Cancer BioResource. Patient data, including anthropometric data, current medication, and familial history, were collected at St Vincent’s Hospital prior to radical prostatectomy and collated by the APCRC-NSW/Australian Prostate Cancer BioResource. PCa pathological analysis and prostate characterization were conducted at Douglass Hanly Moir Pathology (Macquarie Park, NSW Australia). Histopathological data, including lymph node invasion, extracapsular extension, and PCa grading via ISUP and Gleason score (including total, primary, and secondary Gleason score), were provided to and collated by the Australian Prostate Cancer BioResource. Analysis of patient blood serum was undertaken by Royal Prince Alfred Hospital Pathology laboratory, Sydney, NSW, Australia.

The study was approved by the St. Vincent’s Hospital Sydney Research Ethics Committee (12/231).

### 4.2. Adipose Lipolysis

PPAT and SAT explants were dissected into five pieces (~30–50 mg each) and incubated in Kreb’s buffer (125 mM NaCl, 5 mM KCl, 2 mM CaCl_2_, 1.3 mM KH_2_PO_4_, 1.3 mM MgSO_4_·7H_2_O, 24.8 mM NaHCO_3_), 2% fatty acid-free BSA, and 5.5 mM glucose. Explants were treated with either 10 nM insulin (Bovine Insulin, Sigma-Aldrich, Sydney, Australia), 1 μM (maximal) and 10 nM (mid-dose) dl-Isoproterenol (Sigma-Aldrich), and 1 nM dihydrotestosterone (DHT; Sigma-Aldrich) or dH_2_O (control) for 2 h in a 37 °C water bath. Media non-esterified fatty acids levels was determined using commercial kit following manufacturer’s instructions (NEFA-C, WAKO Diagnostics, Richmond, VA, USA). Media fatty acids were determined using gas chromatography, as described below.

### 4.3. Fatty Acid Profile

The basal secreted fatty acid profile from periprostatic and subcutaneous adipose tissues was analyzed by direct transesterification, as previously described [79]. Then, 200 μL of sample was added to 2 mL of methanol:toluene (4:1) containing 0.01% BHT and 200 μL of internal standard, heneicosaenoic acid (0.2 mg/mL, dissolved in toluene), and 200 μL of acetyl chloride was added to each sample whilst vortexing. Samples were heated at 100 °C for 60 min, then cooled in water. Following this, 5 mL potassium carbonate (6%) was added and subjected to centrifugation (10 min, 2000 g, 4 °C). The upper toluene phase of each sample was recovered and stored in GC vials at −40 °C until analysis. These samples were analyzed by flame-ionization gas chromatography (model GC-201A, Shimadzu, Sydney, Australia) using a 50 m × 0.25 mm internal diameter capillary column. Two microliters of extracted sample were auto-injected into the column, and individual fatty acids were identified by comparison with known fatty acid standards (Nu-Chek Prep, Elysian, MN, USA and Sigma-Aldrich, Sydney, Australia) using the Shimadzu analysis software (Class-VP 7.2.1 SP1, Sydney, Australia).

### 4.4. Adipocyte Size

A portion of PPAT and SAT samples were fixed in 10% buffered formalin for 24–48 h at room temperature and then processed for standard paraffin embedding. The remaining portion of the sample was immediately flash-frozen with liquid nitrogen and subsequently kept at −80 °C. All adipose tissue slides were stained with hematoxylin and eosin. Images were acquired at 20× magnification with an Olympus Bx-53 Fluorescent DP-73 Microscope and cellSens Standard software version 1.8 for PC (cellSens Software, Macquarie Park, Australia). Images of three randomized fields were collected, with each field containing a minimum of 60 cells, as previously defined [80]. Adipocyte size was determined using Adiposoft software (Image J, University of Navarra, Pamplona, Spain), with the following parameter adapted from [81]: “minimum diameter” of 10 µm, a “maximum diameter” of 1000 µm, and “microns per pixel” of 0.439.

### 4.5. Statistical Analysis

GraphPad Prism version 8.4.1 for PC (GraphPad Software, La Jolla, California USA) and Microsoft Excel (Microsoft, Redmond, Washington, USA) were used to statistically analyze all data, with *p* ≤ 0.05 considered significant.

For dichotomized analysis (i.e., GS, WC), non-parametric Wilcoxon signed-rank and Mann–Whitney U-tests were used to determine significance. Linear regression models were used to test for relationships between dichotomized groups. Non-parametric Kruskal–Wallis ANOVAs were used to test for main effect, while posthoc Tukey’s multiple comparisons and Dunn’s multiple comparisons were undertaken. Two-way ANOVAs were conducted to test for main effect, while posthoc Tukey’s multiple comparisons were undertaken.

## 5. Conclusions

The data presented show that the lipid biological features of PPAT do differ from SAT; however, we did not observe any meaningful difference in ex vivo PPAT biology that align with differences in PCa aggressiveness or patient metabolic health. Significantly, this is the first study to comprehensively assess PPAT lipid biology and compare it to paired SAT. In combination with other studies that reported that PPAT thickness [42,43], adipose infiltration into the prostate [14], altered fatty acid composition [30,54], inflammation [61], and altered profile of secreted adipokines [21,48] are associated with more aggressive PCa, we propose a new model linking PPAT biology to PCa aggressiveness. In high-grade disease, the PPAT has more immune cells [61], there is an increase secretion of pro-oncogenic adipocytokines [21,48], and the PPAT mass is greater [42,43], and therefore, there is a greater bulk secretion of FAs. This latter observation is also observed in obesity, where there is no difference in basal fatty acid release on a per-milligram basis but there is elevated fatty acid availability due to increased adipose mass [82,83]. Additional studies are required to confirm this model further and thereby identify novel therapeutic targets to uncouple the supportive role of PPAT in PCa progression.

## Figures and Tables

**Figure 1 cancers-12-01385-f001:**
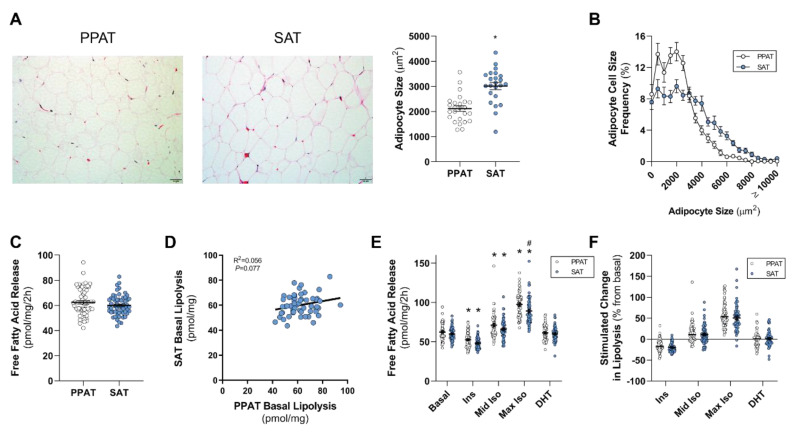
Periprostatic and subcutaneous adipose tissue adipocyte size and lipolysis. (**A**) Representative images and mean and (**B**) distribution of adipocyte size of periprostatic adipose tissue (PPAT) and subcutaneous adipose tissue (SAT); (**C**) basal fatty acid release rates from PPAT and SAT and (**D**) linear regression analysis of PPAT and SAT basal free fatty acid release. (**E**) Absolute free fatty acid release from PPAT and SAT in response to 10 nM insulin (Ins), 10 nM (mid Iso), and 1 µM (max Iso) isoproterenol, and 1 nM dihydrotestosterone (DHT). (**F**) Change in PPAT and SAT fatty acid release from the basal rate in response to 10 nM insulin (Ins), 10 nM (mid Iso), and 1 µM (max Iso) isoproterenol, and 1 nM dihydrotestosterone (DHT). All data were expressed as mean ± SEM. * *p* < 0.05 by a non-parametric Mann–Whitney U-test (A,C,F); * *p* < 0.05 vs. basal (same fat bed), # *p* < 0.05 vs. PPAT by a two-way ANOVA (E). *N* = 20 for A and B, *N* = 60 for C–F. Scale bar: 50 µm.

**Figure 2 cancers-12-01385-f002:**
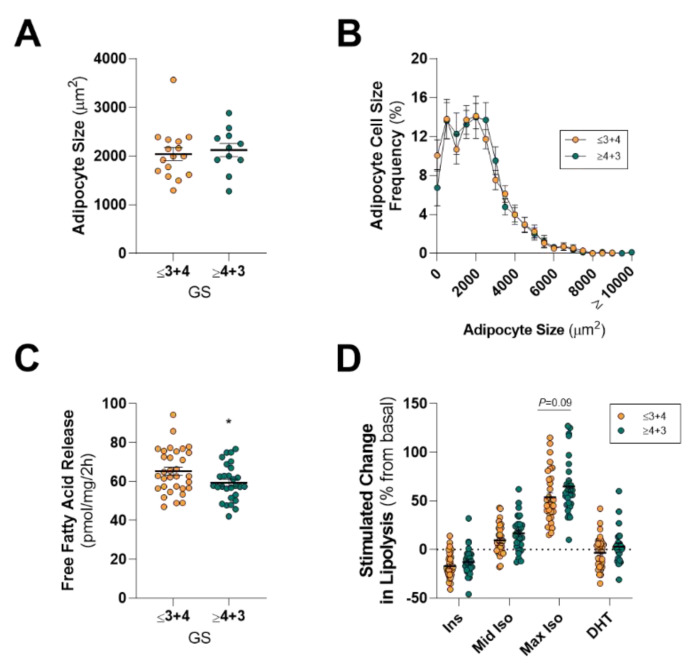
Comparison of periprostatic adipose tissue adipocyte size and lipolysis in men with less aggressive and more aggressive prostate cancer. (**A**) Mean and (**B**) distribution of adipocyte size of periprostatic adipose tissue (PPAT) dichotomized into Gleason score ≤3 + 4 (less aggressive) and ≥4 + 3 (more aggressive) groups; (**C**) Basal fatty acid release rates and (**D**) absolute free fatty acid release from PPAT in response to 10 nM insulin (Ins), 10 nM (mid Iso) and 1 µM (max Iso) isoproterenol, and 1 nM dihydrotestosterone (DHT) dichotomized into Gleason score ≤3 + 4 (less aggressive) and ≥4 + 3 (more aggressive) groups. All data was expressed as mean ± SEM. (**A**,**B**) GS ≤ 3 + 4, n = 16; ≥4 + 3, n = 11. (**C**,**D**) GS ≤ 3 + 4, n = 32; ≥ 4 + 3, *n* = 28. * *p* < 0.05 by a non-parametric Mann–Whitney *U*-test (**C**), * *p* < 0.05 by a two-way ANOVA (**D**).

**Figure 3 cancers-12-01385-f003:**
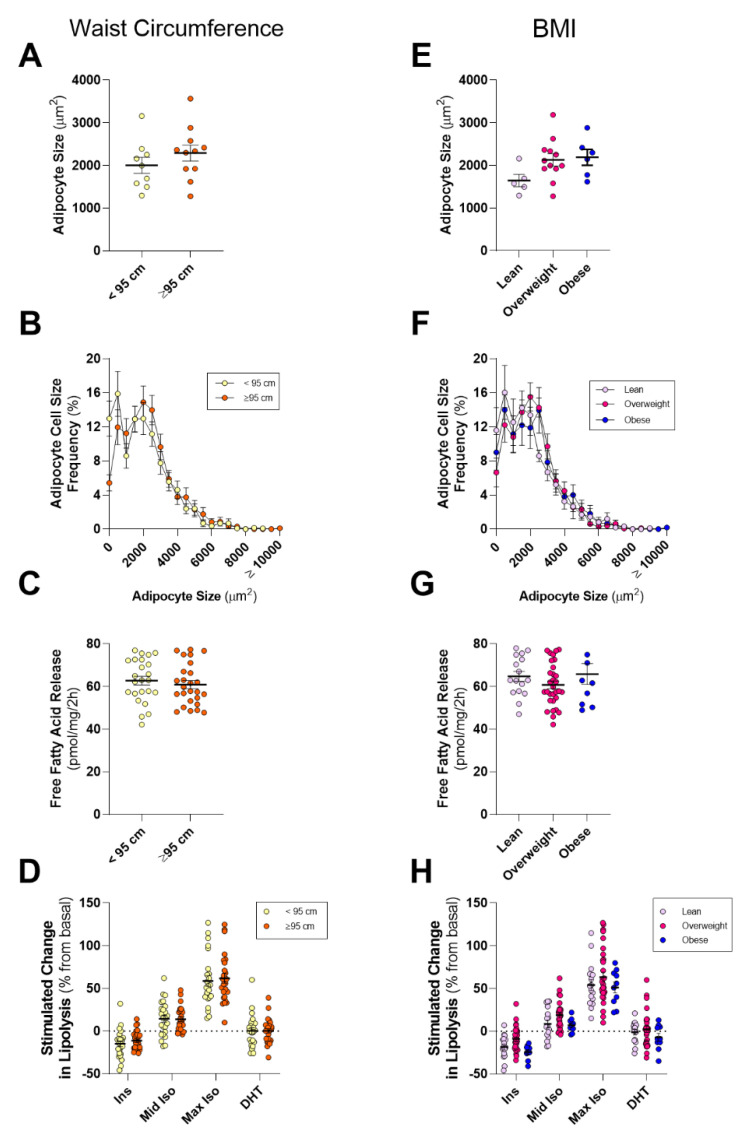
Comparison of periprostatic adipose tissue adipocyte size and lipolysis in lean or obese men with prostate cancer. (**A**) Mean and (**B**) distribution of adipocyte size of periprostatic adipose tissue (PPAT) dichotomized into waist circumference <95 cm (normal) and ≥95 cm (high) groups. (**C**) Basal fatty acid release rates and (**D**) absolute free fatty acid release from PPAT in response to 10 nM insulin (Ins), 10 nM (mid Iso), and 1 µM (max Iso) isoproterenol, and 1 nM dihydrotestosterone (DHT) dichotomized into waist circumference <95 cm (normal) and ≥95 cm (high) groups. (**E**) Mean and (**F**) distribution of adipocyte size of periprostatic adipose tissue (PPAT) stratified by BMI into Lean (<25 kg/m^2^), Overweight (≥25 kg/m^2^, <30 kg/m^2^), and Obese (≥30 kg/m^2^) groups. (**G**) Basal fatty acid release rates and (**H**) absolute free fatty acid release from PPAT in response to 10 nM insulin (Ins), 10 nM (mid Iso) and 1 µM (max Iso) isoproterenol, and 1 nM dihydrotestosterone (DHT) stratified by BMI into Lean, Overweight and Obese groups. PPAT n = 25, ≤95 cm n = 9, ≥95 cm n = 11, Lean n = 7, Overweight n = 12, Obese n = 6. SAT n = 25, ≤95 cm n = 6, ≥95 cm n = 12, Lean n = 7, Overweight n = 11, Obese n = 5. Data are mean ± SEM.

**Table 1 cancers-12-01385-t001:** Summary of the patient baseline characteristics from men undergoing a radical prostatectomy.

Characteristics	Total	Range
Age, years	62 ± 8 (60)	42–78
PSA, µg/L	6.88 ± 5.50 (58)	1.4–30
BMI, kg/m^2^	26.9 ± 3.4 (57)	21.3–35.6
Waist Circumference, cm	95.8 ± 8.8 (50)	81–117
Gleason Score		
≤3 + 3	3	
3 + 4	29	
4 + 3	13	
≥4 + 4	15	
Pathological T Stage		
T2	29	
T3a	27	
T3b	4	
T4	0	
Blood Serum, n	60	
Glucose, mmol/L	5.3 ± 0.7	3.6–7.1
Insulin, pmol/L	31 ± 23	7–136
Cholesterol, mmol/L	4.5 ± 1.0	2.4–6.2
Triacylglycerol, mmol/L	1.2 ± 0.4	0.5–2.4
HDL, mmol/L	1.3 ± 0.4	0.76–2.63
LDL, mmol/L	2.6 ± 1.0	0.3–5.2

Data are mean ± SD (*n*), except for Gleason score and pathological T stage, which is the number of cases.

**Table 2 cancers-12-01385-t002:** The basal secreted fatty acid profile from periprostatic and subcutaneous adipose tissue from men undergoing radical prostatectomy.

Fatty Acid (mol %)	PPAT (*n* = 54)	SAT (*n* = 48)
SFA		
14:0	2.47 ± 0.08	2.26 ± 0.11
15:0	0.07 ± 0.03	0.04 ± 0.02
16:0	22.32 ± 0.18	21.54 ± 0.18 *
17:0	0.84 ± 0.04	0.79 ± 0.04
18:0	15.89 ± 0.31	16.50 ± 0.46
20:0	0.45 ± 0.11	0.19 ± 0.04 *
22:0	0.06 ± 0.02	0.04 ± 0.04
24:0	0.63 ± 0.04	0.70 ± 0.05
MUFA		
14:1 n-9	0.04 ± 0.02	0.05 ± 0.02
15:1 n-10	0.11 ± 0.04	0.30 ± 0.06 *
16:1 n-7	2.46 ± 0.10	2.63 ± 0.14
17:1 n-7	0.02 ± 0.02	0.04 ± 0.02
18:1 n-7	0.67 ± 0.05	0.62 ± 0.05
18:1 n-9	31.16 ± 0.46	30.65 ± 0.61
20:1 n-9	0.20 ± 0.04	0.13 ± 0.03
22:1 n-9	1.71 ± 0.31	0.62 ± 0.14 *
24:1 n-9	0.07 ± 0.02	0.13 ± 0.04
PUFA *n-6*		
18:2 n-6	12.00 ± 0.23	12.20 ± 0.21
18:3 n-6	0.11 ± 0.04	0.32 ± 0.07 *
20:3 n-6	1.28 ± 0.05	1.54 ± 0.08 *
20:4 n-6	1.84 ± 0.06	2.21 ± 0.09 *
22:2 n-6	0.04 ± 0.04	0.45 ± 0.04
22:4 n-6	0.04 ± 0.03	0.10 ± 0.05
22:5 n-6	0.04 ± 0.04	0.13 ± 0.06
22:2 n-6	0.36 ± 0.04	0.45 ± 0.04
PUFA *n-3*		
18:3 n-3	2.24 ± 0.06	2.31 ± 0.07
20:5 n-3	1.02 ± 0.04	1.10 ± 0.05
22:5 n-3	1.64 ± 0.05	1.83 ± 0.05 *
22:6 n-3	0.27 ± 0.04	0.58 ± 0.10 *
Summary statistics		
∑ SFA	42.71 ± 0.42	42.07 ± 0.50
∑ MUFA	36.46 ± 0.52	35.16 ± 0.75
∑ PUFA	20.83± 0.37	22.77 ± 0.5 *
∑ n-6	15.66 ± 0.27	16.95 ± 0.34 *
∑ n-3	5.17 ± 0.15	5.81 ±0.18 *

Mean ± SEM are shown. * *p* < 0.05 by a Student *t*-test. PPAT: periprostatic adipose tissue; SAT: subcutaneous adipose tissue; SFA: saturated fatty acid; MUFA: monounsaturated fatty acid; PUFA: polyunsaturated fatty acids.

**Table 3 cancers-12-01385-t003:** Summary of patient baseline characteristics dichotomized by Gleason score into a less aggressive “≤3 + 4” cohort and more aggressive “≥4 + 3” cohort.

Characteristics	≤3 + 4 (*n* = 32)	≥4 + 3 (*n* = 28)	*p*
Age, years	58 ± 8	65 ± 7	0.006
PSA, µg/L	6.04 ± 5.21 (30)	7.79 ± 5.65 (28)	0.01
BMI, kg/m^2^	26.8 ± 3.5 (32)	27.0 ± 3.3 (25)	0.70
Waist Circumference, cm	93.7 ± 8.9 (26)	98.1 ± 8.1 (24)	0.054
Pathological T Stage			
T2	19	10	
T3a	13	14	
T3b	1	3	
T4	0	0	
Blood Serum			
Glucose, mmol/L	5.2 ± 0.6	5.4 ± 0.8	0.34
Insulin, pmol/L	29 ± 19	33 ± 26	0.72
Cholesterol, mmol/L	4.6 ± 1.0	4.5 ± 0.9	0.61
Triacylglycerols, mmol/L	1.3 ± 0.5	1.2 ± 0.3	0.82
HDL, mmol/L	1.3 ± 0.4	1.3 ± 0.4	0.56
LDL, mmol/L	2.6 ± 1.0	2.6 ± 0.8	0.81

Mean ± SD (n) except for the pathological T stage, which is the number of cases.

**Table 4 cancers-12-01385-t004:** The basal secreted fatty acid profile of PPAT dichotomized by Gleason score into a less aggressive “≤3 + 4” cohort and a more aggressive “≥4 + 3” cohort.

Fatty Acid (mol %)	≤3 + 4 (*n* = 29)	≥4 + 3 (*n* = 25)
SFA		
14:0	2.58 ± 0.10	2.35 ± 0.12
15:0	0.11 ± 0.04	0.04 ± 0.02
16:0	22.09 ± 0.28	22.58 ± 0.19
17:0	0.83 ± 0.06	0.85 ± 0.06
18:0	15.76 ± 0.45	16.04 ± 0.43
20:0	0.63 ± 0.18	0.23 ± 0.11
22:0	0.07 ± 0.03	0.04 ± 0.03
24:0	0.60 ± 0.06	0.66 ± 0.06
MUFA		
14:1 n-9	0.02 ± 0.02	0.06 ± 0.04
15:1 n-10	0.07 ± 0.05	0.17 ± 0.07
16:1 n-7	2.39 ± 0.12	2.54 ± 0.18
17:1 n-7	0.03 ± 0.03	0.01 ± 0.01
18:1 n-7	0.66 ± 0.07	0.69 ± 0.06
18:1 n-9	31.52 ± 0.62	30.75 ± 0.68
20:1 n-9	0.17 ± 0.05	0.24 ± 0.06
22:1 n-9	1.84 ± 0.43	1.55 ± 0.47
24:1 n-9	0.11 ± 0.03	0.03 ± 0.02 *
PUFA *n-6*		
18:2 n-6	11.80 ± 0.34	12.24 ± 0.30
18:3 n-6	0.14 ± 0.06	0.06 ± 0.03
20:3 n-6	1.29 ± 0.08	1.28 ± 0.07
20:4 n-6	1.77 ± 0.11	1.92 ± 0.06
22:2 n-6	0.36 ± 0.05	0.35 ± 0.05
22:4 n-6	0.07 ± 0.05	ND
22:5 n-6	0.08 ± 0.08	ND
PUFA *n-3*		
18:3 n-3	2.17 ± 0.10	2.32 ± 0.06
20:5 n-3	1.01 ± 0.05	1.05 ± 0.06
22:5 n-3	1.56 ± 0.08	1.73 ± 0.05
22:6 n-3	0.27 ± 0.05	0.26 ± 0.06
Summary statistics		
∑ SFA	42.67 ± 0.64	42.77 ± 0.55
∑ MUFA	36.83 ± 0.73	36.03 ± 0.73
∑ PUFA	20.50 ± 0.59	21.21 ± 0.41
∑ n-6	14.98 ± 0.65	15.85 ± 0.32
∑ n-3	4.84 ± 0.29	5.36 ± 0.15

Mean ± SEM are shown. * *p* < 0.05 by a Student *t*-test. SFA: saturated fatty acid; MUFA: monounsaturated fatty acid; PUFA: polyunsaturated fatty acid.

**Table 5 cancers-12-01385-t005:** Summary of patient baseline characteristics dichotomized by waist circumference.

Characteristics	Normal (<95 cm; *n* = 24)	High (≥95 cm; *n* = 26)	*p* *
Age, years	61 ± 9	64 ± 6	0.46
PSA, µg/L	6.98 ± 6.85 (23)	7.24 ± 4.93 (26)	0.74
BMI, kg/m^2^	25.12 ± 1.98 (24)	28.37 ± 3.53 (26)	0.0005
Waist Circumference, cm	88.65 ± 4.37 (24)	102.46 ± 6.23 (26)	<0.0001
Blood Serum			
Glucose, mmol/L	5.3 ± 0.7	5.2 ± 0.8	0.97
Insulin, pmol/L	24 ± 11	37 ± 27	0.13
Cholesterol, mmol/L	4.3 ± 0.8	4.7 ± 0.9	0.04
Triacylglycerols, mmol/L	1.2 ± 0.4	1.3 ± 0.4	0.80
HDL, mmol/L	1.3 ± 0.4	1.4 ± 0.4	0.25
LDL, mmol/L	2.4 ± 0.8	2.7 ± 0.9	0.07

Mean ± SD are shown. * *p* < 0.05 by a Student *t*-test.

**Table 6 cancers-12-01385-t006:** The basal secreted fatty acid profile of PPAT classified by waist circumference and BMI.

Fatty Acid (mol %)	WC		BMI		
	Normal (<95 cm) (*n* = 23)	High (≥95 cm) (*n* = 22)	Lean (*n* = 15)	Overweight (*n* = 28)	Obese (*n* = 9)
SFA					
14:0	2.62 ± 0.10	2.37 ± 0.08	2.46 ± 0.13	2.49 ± 0.08	2.65 ± 0.17
15:0	0.11 ± 0.05	0.05 ± 0.04	0.10 ± 0.05	0.08 ± 0.04	ND
16:0	22.2 ± 0.34	22.35 ± 0.19	21.65 ± 0.44	22.44 ± 0.20	22.89 ± 0.33
17:0	0.94 ± 0.04	0.83 ± 0.06	0.86 ± 0.08	0.88 ± 0.04	0.74 ± 0.15
18:0	15.77 ± 0.51	15.94 ± 0.46	15.54 ± 0.56	15.72 ± 0.41	16.43 ± 0.95
20:0	0.35 ± 0.09	0.36 ± 0.14	0.27 ± 0.09	0.45 ± 0.12	0.83 ± 0.54
22:0	0.05 ± 0.02	0.08 ± 0.04	0.06 ± 0.03	0.08 ± 0.03	ND
24:0	0.65 ± 0.04	0.64 ± 0.06	0.61 ± 0.07	0.66 ± 0.04	0.54 ± 0.11
MUFA					
14:1 n-9	0.07 ± 0.03	0.03 ± 0.03	0.05 ± 0.05	0.05 ± 0.04	ND
15:1 n-10	0.08 ± 0.06	0.14 ± 0.06	0.12 ± 0.09	0.14 ± 0.09	0.05 ± 0.05
16:1 n-7	2.46 ± 0.14	2.52 ± 0.20	2.39 ± 0.17	2.55 ± 0.17	2.40 ± 0.23
17:1 n-7	0.06 ±0.04	ND	0.02 ± 0.02	0.03 ± 0.03	ND
18:1 n-7	0.77 ± 0.06	0.67 ± 0.06	0.68 ± 0.10	0.73 ± 0.04	0.54 ± 0.14
18:1 n-9	31.73 ± 0.65	30.58 ± 0.66	32.26 ± 0.85	30.88 ± 0.56	30.65 ± 1.41
20:1 n-9	0.18 ± 0.06	0.27 ± 0.06	0.17 ± 0.07	0.28 ± 0.07	0.06 ± 0.06
22:1 n-9	1.13 ± 0.25	2.08 ± 0.59	1.04 ± 0.33	1.84 ± 0.50	2.23 ± 0.90
24:1 n-9	0.10 ± 0.04	0.06 ± 0.03	0.10 ± 0.04	0.09 ± 0.3	ND
PUFA *n-6*					
18:2 n-6	11.82 ± 0.35	11.85 ± 0.31	12.66 ± 0.50	11.68 ± 0.28	11.63 ± 0.47
18:3 n-6	0.17 ± 0.08	0.07 ± 0.07	0.14 ± 0.05	0.13 ± 0.07	ND
20:3 n-6	1.32 ± 0.08	1.35 ± 0.05	1.32 ± 0.06	1.31 ± 0.07	1.26 ± 0.18
20:4 n-6	1.73 ± 0.08	2.02 ± 0.08 *****	1.84 ± 0.08	1.80 ± 0.07	1.92 ± 0.29
22:2 n-6	0.37 ± 0.05	0.40 ± 0.05	0.35 ± 0.07	0.40 ± 0.05	0.32 ± 0.11
22:4 n-6	0.06 ± 0.06	0.03 ± 0.03	ND	0.07 ± 0.05	ND
22:5 n-6	0.10 ± 0.04	ND	ND	0.08 ± 0.08	ND
PUFA *n-3*					
18:3 n-3	2.23 ± 0.07	2.25 ± 0.07	2.32 ± 0.07	2.21 ± 0.06	2.07 ± 0.28
20:5 n-3	1.02 ± 0.04	1.09 ± 0.04	1.05 ± 0.05	1.03 ± 0.04	1.03 ± 0.15
22:5 n-3	1.59 ± 0.06	1.71 ± 0.06	1.65 ± 0.07	1.61 ± 0.05	1.60 ± 0.24
22:6 n-3	0.32 ± 0.06	0.26 ± 0.06	0.29 ± 0.07	0.31 ± 0.05	0.16 ± 0.11
Summary statistics					
∑ SFA	42.68 ± 0.79	42.62 ± 0.45	41.55 ± 0.89	42.80 ± 0.58	44.08 ± 0.50
∑ MUFA	36.57 ± 0.78	36.35 ± 0.81	36.83 ± 1.03	36.58 ± 0.66	35.93 ± 1.54
∑ PUFA	20.74 ± 0.47	21.01 ± 0.55	21.62 ± 0.59	20.62 ± 0.47	19.98 ± 1.36
∑ n-6	14.90 ± 0.77	15.70 ± 0.40	16.30 ± 0.51	15.46 ± 0.34	15.13 ± 0.79
∑ n-3	4.94 ± 0.28	5.32 ± 0.18	5.32 ± 00.18	5.16 ± 0.34	4.86 ± 0.67

Mean ± SEM are shown. * *p* < 0.05 by a Student *t*-test. SFA: saturated fatty acid; MUFA: monounsaturated fatty acid; PUFA: polyunsaturated fatty acids.

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
