# Peer review of "Assessment of Periprostatic and Subcutaneous Adipose Tissue Lipolysis and Adipocyte Size from Men with Localized Prostate Cancer"

_cancers, 2020, doi:10.3390/cancers12061385_

Round 1

Reviewer 1 Report

This interesting study is about the prostatic periprostatic adipose tissue (PPAT) to discuss whether thickness associated with more aggressive prostate cancer. The authors selected PPAT and SAT from 60 men undergoing total prostatectomy for prostate cancer. They found that basal lipolysis of PPAT was increased in men diagnosed with less aggressive PCa compared to men with more aggressive PCa (GS ≥ 4+3). It’s an interesting study with robust research data proven their hypotheses. There are some concerns in this study and need the authors to address.

Major Concerns:

  • The authors defined high aggressive of prostate cancer according Gleason score and divided them into less aggressive “GS ≤ 3+4” and a more aggressive group “GS ≥ 4+3” since patients with “≥ 4+3” disease. However , GS is only one part of the stages of prostate cancer. It is not appropriate to define prostate cancer without staging TNM system. I suggested category with TNM system.
  • In all 60 patients enrolled in this study, there existed outliner and should be excluded to reduce the bias e.g. the patient with BMI=35.6 WC=117 etc.
  • For this study did not discuss signal transduction or inflammation these important factors of PCa progression, the result seemed not to support the conclusion of PPAT may not associated with PCa progression.

Minor comments:

  • The authors need to edit this paper for professionally English language.

Author Response

We thank the reviewer for their comments and suggestions.

1. The authors defined high aggressive of prostate cancer according Gleason score and divided them into less aggressive “GS ≤ 3+4” and a more aggressive group “GS ≥ 4+3” since patients with “≥ 4+3” disease. However, GS is only one part of the stages of prostate cancer. It is not appropriate to define prostate cancer without staging TNM system. I suggested category with TNM system. 

Response – we agree with the reviewer that Gleason score is one criterion in the clinical definitions of high, intermediate and low risk disease, which also incorporate pathological stage (TNM) and preoperative PSA levelsWe have now added pathological stage information to Tables 1 and 3, alongside Gleason Score and PSA. Of specific relevance to this study, our team’s published analyses demonstrate that Gleason Score is the most powerful predictor of outcome in the era of PSA testing, in that lower PSA levels and lower stage of cancer (Grogan et al, BJU Int. 2017 Nov;120(5):651-658, PMID: 28371244; Thanigasalam et al, BJU Int. 2010 Mar;105(5):642-7, PMID: 19751263). 

2. In all 60 patients enrolled in this study, there existed outliner and should be excluded to reduce the bias e.g. the patient with BMI=35.6 WC=117 etc.  

Response –Our cohort contained 4 men with WC >110 cm, and 9 men with a BMI >30, and no outliers were determined using the Grubbs Test (extreme studentized deviate method). 

3. For this study did not discuss signal transduction or inflammation these important factors of PCa progression, the result seemed not to support the conclusion of PPAT may not associated with PCa progression. 

Response –We agree that inflammation in particular is potentially important in PPAT biology, indeed there have been eight previous publications describing PPAT inflammation and prostate cancer; however, there has not been an analysis of the lipid biology of PPAT. Therefore, our study aimed to determine whether ex vivo attributes of adipocyte lipid biology differed in men with low or high-grade PCa or obesityIn our Conclusions (lines 434-447), wdiscuss other aspects of PPAT biology (inflammation, adipokines etc) and their links to PCa progression. 

Reviewer 2 Report

The hypothesis seems scientifically sound but very obscure in nature

The pre prostate adipose tissue is no more a part of the prostate gland than the peri rectal or pelvic adipose tissue. In fact the prostate blood supply and lymphatics are devoid in the PPAT area and therefore not part of prostate tumour biology. 

Its always a concern when the conclusion proposes another study with a similar hypothesis

Author Response

We thank the reviewer for their comments and suggestions.

1) The pre prostate adipose tissue is no more a part of the prostate gland than the peri rectal or pelvic adipose tissue. In fact the prostate blood supply and lymphatics are devoid in the PPAT area and therefore not part of prostate tumour biology.  

Response – we thank the reviewer for their insights. We do not propose that PPAT is a part of the prostate gland, only that PPAT surrounds prostate, which is a statement supported by ~70 publications (number of PubMed results for “periprostatic adipose tissue”)Further, we recently reviewed this extensive literature on PPAT including a substantial section on the various imaging techniques used to assess PPAT thickness (Nassar et al, BJU Int 2018, 121 Suppl 3, 9-21). More recently, Estève and colleagues, which included Drs Rouminguié and Manceau from the Département d’Urologie, CHU de Toulouse (Current Opinion in Endocrine and Metabolic Research 2020, 10, 29-35) provided additional evidence of the close proximity of PPAT to the prostate (including original MRI sections) and that “vascularization of the prostate crosses PPAT”. Both of these reviews of the literature provide published evidence that PPAT is anatomically closely located to the prostate and correlates with prostate cancer features. Further, both reviews were co-authored by surgeons, and summarise many publications authored by clinical teams. 

2) Its always a concern when the conclusion proposes another study with a similar hypothesis 

Response – Our conclusion is that ex vivo aspects of PPAT lipid biology are not strikingly different across patients with differing obesity profiles and disease aggressiveness we detected no evidence of ‘metabolic remodellingHowever, our observations do not negate the possibility that the in vivo milieu, influenced by local and systemic signals including tumour-derived signals, will result in altered lipolytic flux and PPAT biologyAs such, we propose additional studies that do attempt to assess in vivo biology of PPAT in men with low or high aggressive prostate cancer. 

Reviewer 3 Report

The Authors have investigated a current topic of extreme interest in patients with prostate cancer. The paper is well structured and well written.

I suggest to assess potential/future implications of this model into the clinical practice and to stress a bit more the weaknesses in an attempt to positively stimulate the attention of the Readers. Again, along this line which directions of research could be explored? I think the Authors should indicate this.

Just a few points to review/improve:     

1. I suggest to implement the keywords introducing the items: "localized prostate cancer" and "subcutaneous adipose tissue"

2. Into Discussion, lines 256-257 and 262-263, I believe there was a writing position "mistake": high-grade PCa disease (GS<3+4)? low-grade disease (GS>4+3)?

Author Response

We thank the reviewer for their comments and suggestions.

I suggest to assess potential/future implications of this model into the clinical practice and to stress a bit more the weaknesses in an attempt to positively stimulate the attention of the Readers. Again, along this line which directions of research could be explored? I think the Authors should indicate this.

Response  we believe that we have proposed the limitations of our study and the remaining issues of this area of research throughout the discussion. For example, in the Discussion (line 347-350) states the likelihood that in vivo responses will almost certainly differ to our in vitro/ex vivo measures, which is expanded in the Conclusions (line 435-448) where we integrate the existing literature and state the need for additional experiments to address these limitations. Additionally, we highlight the complex and unresolved issues related to understanding the relationship between obesity, PPAT thickness and prostate cancer biology.

Just a few points to review/improve:     

1. I suggest to implement the keywords introducing the items: "localized prostate cancer" and "subcutaneous adipose tissue"

Response  thank you for the suggestions, we have added these terms.

2. Into Discussion, lines 256-257 and 262-263, I believe there was a writing position "mistake": high-grade PCa disease (GS<3+4)? low-grade disease (GS>4+3)?

Response  thank you for identifying this mistake. This has been corrected on lines 263-4

Reviewer 4 Report

The information that is presented in this manuscript demonstrates that the lipid biological features of periprostatic adipose tissue (PPAT) do differ from subcutaneous adipose tissue (SAT). Concretely, PPAT adipocytes are smaller and secrete less Poly-Unsaturated Fatty Acids (PUFA) species than SAT. However, no relevant difference in ex vivo PPAT biology that associated with prostate cancer aggressiveness or obesity was observed.

Overall, the experiments are well-designed, well-controlled, and support the conclusions that are made by the authors although I think that molecular assays would have contributed to improve the quality of the present study (for example, studying levels of TNF alpha or VEGF of which overexpression is a marker of aggressiveness of prostate cancer). On the other hand, the results are clearly presented and the discussion is well written, although perhaps it is quite long.

General comments

As is noted in detail below, there are a few concerns that should be addressed prior to this paper is considered for publication:

1. Some of the results shown in this manuscript have already been published previously, although I cannot check if that is a dissertation or something like that. Could you clarify it?

2. Did you find any difference in the aggresiveness of the prostate cancer between men taking statins compared to men not taking statins?

3. In general, the quality of the figures should be improved. The text is blurry (see, for example the next comment).

4. Although the abstract says “Adipocites in PPAT were less responsive to insulin-stimulated suppression of lipolysis (line 33)” and the discussion says “PPAT lipolysis responded as expected to insulin (anti-lipolytic) and isoproterenol (pro-lipolytic); however, PPAT was slightly less responsive compared to SAT explants”, it is not explained in the results (lines 111-112). I cannot see these differences between PPAT and SAT in the figure 1E (or if they are significant), because the quality of the figure is low, so please, review it.

Other comments

Line 123. Delete “mean and”

Line 127. “Change” must be written with a capital letter.

Line 135. the abbreviations PPAT and SAT must be explained in the legends of figures.

In lines 256-263 there is a mistake. In these lines high-grade PCa disease is defined as GS ≤ 3+4, and low-grade disease is defined as GS ≥ 4+3.

Line 358. Change “trended” by “trend”

Line 400 and 401. Change 200ul by 200 μl.

Line 402. Change “was” by “were”

Author Response

We thank the reviewer for their comments and suggestions.

1. Some of the results shown in this manuscript have already been published previously, although I cannot check if that is a dissertation or something like that. Could you clarify it? 

Response – These results have not been previously published. This study was the basis of a Bachelor of Science (Hons) thesis and presented as a poster at a local Australian conference in 2018.. 

2. Did you find any difference in the aggresiveness of the prostate cancer between men taking statins compared to men not taking statins? 

Response – We only had 9 Men taking statins in our cohort of 60. The Gleason Score distribution is in the table below. 

Entire Cohort 

No Statin 

Statin  

GS 

N 

% 

N 

% 

N 

% 

≤ 3+3 

3 

5% 

3 

6% 

0 

0% 

3+4 

29 

48% 

23 

45% 

6 

67% 

4+3 

13 

22% 

11 

22% 

2 

22% 

≥ 4+4 

15 

25% 

14 

27% 

1 

11% 

3. In general, the quality of the figures should be improved. The text is blurry (see, for example the next comment). 

Response – The low-quality figures in the manuscript were evident only after submission to the journal. We uploaded a MS Word file with high-quality 300 dpi .png figures, and we were not aware that this loss of image quality had occurred.  

4. Although the abstract says “Adipocites in PPAT were less responsive to insulin-stimulated suppression of lipolysis (line 33)” and the discussion says “PPAT lipolysis responded as expected to insulin (anti-lipolytic) and isoproterenol (pro-lipolytic); however, PPAT was slightly less responsive compared to SAT explants”, it is not explained in the results (lines 111-112). I cannot see these differences between PPAT and SAT in the figure 1E (or if they are significant), because the quality of the figure is low, so please, review it. 

Response – Apologies for the confusion and we thank the reviewer for identifying this conflict. We have corrected the abstract (line 33) and the discussion (line 243-44) to reflect our observations. 

Other comments 

Line 123. Delete “mean and 

Response – Figure 1A includes representative images and the mean adipocyte size, Figure 1B is the distribution of adipocyte size. 

Line 127. “Change” must be written with a capital letter. 

Response – this is corrected on line 127 

Line 135. the abbreviations PPAT and SAT must be explained in the legends of figures. 

Response – this has been added to lines 134-5  

In lines 256-263 there is a mistake. In these lines high-grade PCa disease is defined as GS ≤ 3+4, and low-grade disease is defined as GS ≥ 4+3. 

Response – this has been corrected on lines 263-4 

Line 358. Change “trended” by “trend” 

Response – this has been corrected on lines 359 

Line 400 and 401. Change 200ul by 200 μl. 

Response – this has been corrected on lines 401-3 

Line 402. Change “was” by “were” 

Response – we cannot identify the issue that needs resolving. 

Round 2

Reviewer 1 Report

Thank you very much for detail reply the questions. This critical step in development of a biomarker , Gleason score, and detail TNM stage are required for confidence in the findings. As such there is no practical use for the data as they currently stand. Unfortunately I think there will be limited interest in the study as a result.

Thank you for your interest in the Cancers, and we hope you will consider submitting manuscripts to us again in the future.

Reviewer 4 Report

The authors have addressed the changes which were previously suggested. The experiments are well-designed, well-controlled and the results support the conclusions that are made by the authors.

I hope the figures have high quality when the article is published.